# VISION-RWKV: EFFICIENT AND SCALABLE VISUAL PERCEPTION WITH RWKV-LIKE ARCHITECTURES

**Yuchen Duan**[1,2*]**, Weiyun Wang**[3,2*]**, Zhe Chen**[4,2*]**, Xizhou Zhu**[5,2,6]**, Lewei Lu**[6]**,
Tong Lu**[4]**, Yu Qiao**[2]**, Hongsheng Li**[1]**, Jifeng Dai**[5,2]**, Wenhai Wang**[1,2✉]
[1]The Chinese University of Hong Kong, [2]Shanghai AI Laboratory, [3]Fudan University,
[4]Nanjing University, [5]Tsinghua University, [6]SenseTime Research

## ABSTRACT

Transformers have revolutionized computer vision and natural language processing, but their high computational complexity limits their application in high-resolution image processing and long-context analysis. This paper introduces Vision-RWKV (VRWKV), a model that builds upon the RWKV architecture from the NLP field with key modifications tailored specifically for vision tasks. Similar to the Vision Transformer (ViT), our model demonstrates robust global processing capabilities, efficiently handles sparse inputs like masked images, and can scale up to accommodate both large-scale parameters and extensive datasets. Its distinctive advantage is its reduced spatial aggregation complexity, enabling seamless processing of high-resolution images without the need for window operations. Our evaluations demonstrate that VRWKV surpasses ViT's performance in image classification and has significantly faster speeds and lower memory usage processing high-resolution inputs. In dense prediction tasks, it outperforms window-based models, maintaining comparable speeds. These results highlight VRWKV's potential as a more efficient alternative for visual perception tasks. Code and models are available at https://github.com/OpenGVLab/Vision-RWKV.

## 1 INTRODUCTION

Vision Transformers (ViTs) (Dosovitskiy et al., 2020; Touvron et al., 2021a; Vaswani et al., 2017; Steiner et al., 2021; He et al., 2021), renowned for their flexibility and global information processing capabilities, have established new benchmarks in a variety of vision tasks in the past few years. However, the quadratic computational complexity associated with ViTs limits their ability to efficiently process high-resolution images and lengthy sequences, posing a significant barrier to their broader application. As a result, the exploration of a visual architecture that integrates the versatility and comprehensive processing strengths of ViTs, while reducing computational demands, has emerged as a crucial area of research.

In recent developments within natural language processing (NLP), models with linear feature aggregation (or called "linear attention") mechanisms like RWKV (Peng et al., 2023) and Mamba (Gu & Dao, 2023) have emerged as popular solutions for achieving heightened efficiency and processing lengthy texts. These innovative models have demonstrated attributes similar to transformers (Devlin et al., 2018; Raffel et al., 2019; Smith et al., 2022b; Liu et al., 2019; Radford et al., 2018; 2019; Brown et al., 2020; Lewis et al., 2019) in NLP tasks, including the ability to handle long-range dependencies and parallel processing. Furthermore, they have also proven to be scalable, performing well with large-scale NLP datasets. Expanding these techniques into the visual domain shows promise in addressing the computational cost challenge encountered by ViTs.

To develop a vision model incorporating a linear attention mechanism based on the aforementioned methods, while ensuring high capacity for large-scale image data and diverse visual tasks, several critical issues need to be addressed. Firstly, the design of spatial feature aggregation operations needs to be reconsidered, taking into account the differences between image and text modalities. For instance, a redesign of kernels and rewriting at the CUDA level are necessary for attention

---

*Equal contribution. ✉Corresponding authors: wangwenhai@pjlab.org.cn

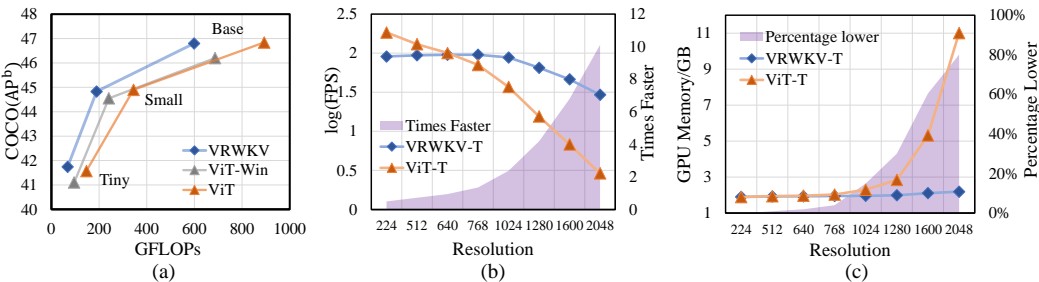

Figure 1: **Performance and efficiency comparison of Vision-RWKV (VRWKV) and ViT.** (a) Bounding box average precision (AP$^b$) comparison of VRWKV and ViT (Touvron et al., 2021a) with window attention and global attention on the COCO (Lin et al., 2014) dataset. (b) Inference speed comparison of VRWKV-T and ViT-T across input resolutions ranging from 224 to 2048. (c) GPU memory comparison of VRWKV-T and ViT-T across input resolutions from 224 to 2048.

mechanisms with a causal receptive field in models like RWKV. Furthermore, the issues of gradient vanishing or exploding tend to arise gradually as the model scales up, resulting in unstable training with large parameter sizes and large-scale datasets. For example, Vision Mamba (Zhu et al., 2024) only gave appropriate results on models with less than 30M parameters. It is important to conduct an in-depth study of how linear attention models can be applied to vision tasks effectively, including examining the scalability concerning data and parameters, assessing the efficiency in handling sparse visual data, and implementing necessary techniques to ensure model stability during scaling up.

Based on these points, we introduce Vision-RWKV (VRWKV). Our approach preserves the core structure and benefits of RWKV (Peng et al., 2023) while incorporating essential changes to process visual data efficiently. Specifically, we design a quad-directional shift (Q-Shift) tailed for vision tasks and modify the original causal RWKV attention mechanism to a bidirectional global attention mechanism (Bi-WKV). The Q-Shift operation expands the semantic range of individual tokens, while the Bi-WKV enables the calculation of global attention within linear complexity in an RNN-form forward and backward. We primarily modify the exponent in the RWKV attention, releasing the limitations of the decay vector and transforming the absolute positional bias into a relative bias. These changes enhance the model's capability while ensuring scalability and stability. In this way, our VRWKV inherits the efficiency of RWKV in handling global information and sparse inputs, while also being able to model the local concept of vision tasks. Additionally, due to severe instability encountered when scaling up the model, we explored a series of measures (Touvron et al., 2021b; Ba et al., 2016) to stabilize the model's outputs. These adjustments significantly improve the model's training stability when scaling up to a larger size.

Building on the aforementioned design, we develop a range of VRWKV models with different model scales, spanning from the VRWKV-Tiny (6M) to the VRWKV-Large (335M). These models are trained using large-scale datasets such as ImageNet-1K (Deng et al., 2009) and ImageNet-22K (Deng et al., 2009). We train them using both common supervised classification and sparse input method MAE (He et al., 2021) and evaluate their performance on visual perception tasks, including classification, detection, and segmentation. Under the same settings, VRWKV has comparable performance to ViT in these tasks with lower computational costs while maintaining stable scalability. This achievement enables VRWKV training parallelism, high flexibility, excellent performance, and low inference cost simultaneously, making it a promising alternative to ViT in a wide range of vision tasks, particularly in high-resolution scenarios.

In this paper, our main contributions are:

(1) We propose VRWKV as a cost-effective alternative to ViT, offering a comprehensive substitute with lower computational requirements. Our model retains ViT's strengths, such as capturing long-range dependencies and handling sparse inputs flexibly, while reducing complexity to a linear scale. This reduction eliminates the need for window operation when processing high-resolution images, making VRWKV a more efficient and scalable solution for vision tasks.

(2) To support vision tasks, we develop a bidirectional global attention mechanism combined with a novel token shift method, Q-Shift, to achieve linear complexity in global attention. Additionally, we implement a set of tailored strategies—integrating relative positional bias, layer scale, and extra layer normalization—to tackle overflow issues and ensure stable, scalable training.

(3) Our model surpasses window-based ViTs and is comparable to global attention ViTs, demonstrating lower FLOPs and GPU memory cost with faster processing speeds as resolution increases, as shown in Figure 1. Notably, VRWKV-T achieves 75.1% top-1 accuracy trained only on the ImageNet-1K (Deng et al., 2009), outperforming DeiT-T (Touvron et al., 2021a) by 2.9 points. With large-scale parameters (*i.e.*, 335M) and training data (*i.e.*, ImageNet-22K), the top-1 accuracy of VRWKV-L is further boosted to 86.0%, which is higher than ViT-L (Dosovitskiy et al., 2020) (86.04 *vs* 85.15). In addition, on COCO (Lin et al., 2014), a challenging downstream benchmark, our best model VRWKV-L achieves 50.6% box mAP, 1.9 points better than ViT-L (50.6 *vs* 48.7).

## 2 RELATED WORKS

### 2.1 VISION ENCODER

Recent advances in vision encoders have significantly pushed the boundaries of computer vision, demonstrating remarkable performance across a range of tasks. Convolutional neural networks (CNNs) served as the foundational model in computer vision. The advancement of computational resources, such as GPUs, has enabled the successful training of stacked convolutional blocks like AlexNet (Krizhevsky et al., 2012) and VGG (Simonyan & Zisserman, 2014) on large-scale image classification datasets (*e.g.*, ImageNet (Deng et al., 2009)). This development paved the way for deeper and more sophisticated convolutional neural architectures, including GoogleNet (Szegedy et al., 2015), ResNet (He et al., 2016), and DenseNet (Huang et al., 2017).

In addition to these innovations, significant advancements have also been made with architectures like SENet (Hu et al., 2018), which introduced a channel attention mechanism to enhance model sensitivity to informative features. Similarly, SKNet (Li et al., 2019) merged multiple kernel sizes to adjust the receptive field adaptively. Further extending the CNN paradigm, recent models such as RepLKNet (Ding et al., 2022) and ConvNeXt (Liu et al., 2022) have refined the convolutional layers to improve efficiency and accuracy, while InternImage (Wang et al., 2023b) explored the strategies to scale up the convolution-based vision model.

Inspired by the success of self-attention layers and transformer architectures in the NLP field, the Vision Transformer (ViT) (Dosovitskiy et al., 2020) applied a transformer framework on image patches, offering a global receptive field and dynamic spatial aggregation. Due to the quadratically computational complexity of the vanilla attention mechanism, approaches like PVT (Wang et al., 2021; 2022) and Linformer (Wang et al., 2020) implemented global attention on down-sampled feature maps, whereas other approaches like Swin (Wu et al., 2022) and HaloNet (Vaswani et al., 2021; Dai et al., 2022) introduced sampling techniques to enlarge the receptive field. Mini-InternVL (Gao et al., 2024) reduces the parameter size of ViT by employing knowledge distillation from a larger ViT to a smaller one, thereby achieving efficiency in the visual encoder.

Another research direction involved replacing self-attention layers in models with linear complexity layers. Representative works include LongNet (Ding et al., 2023), RWKV (Peng et al., 2023), RetNet (Sun et al., 2023), and Mamba (Gu & Dao, 2023), though few have concentrated on visual applications. Concurrently, attempts like Vim (Zhu et al., 2024) and VMamba (Liu et al., 2024) have sought to integrate these linear attention layers into the vision domain. However, these endeavors have only been experimented with on small-scale models (parameters $< 30M$ for Vim and $< 100M$ for VMamba), leaving it uncertain whether their effectiveness extends to larger models.

### 2.2 FEATURE AGGREGATION MECHANISM

The research on feature aggregation has received significant attention. For visual data processing, convolutional operators (LeCun et al., 1995), known for their parameter sharing and local perception, enabled efficient handling of large-scale data through sliding computation. Despite their advantages, traditional CNN operators faced challenges in modeling long-range dependencies. To overcome this issue, advanced convolutional operators, such as the deformable convolution (Dai et al., 2017; Zhu et al., 2019; Xiong et al., 2024), have improved the flexibility of CNN operators, enhancing their long-range modeling capability.

As for the field of NLP, RNN-based operators (Elman, 1990; Memory, 2010; Qin et al., 2024) have historically dominated because of their effectiveness in sequence modeling. RNNs and LSTMs excel in capturing temporal dependencies, making them suitable for tasks requiring an understanding of sequence dynamics. Subsequently, a significant shift occurred. The introduction of the transformer

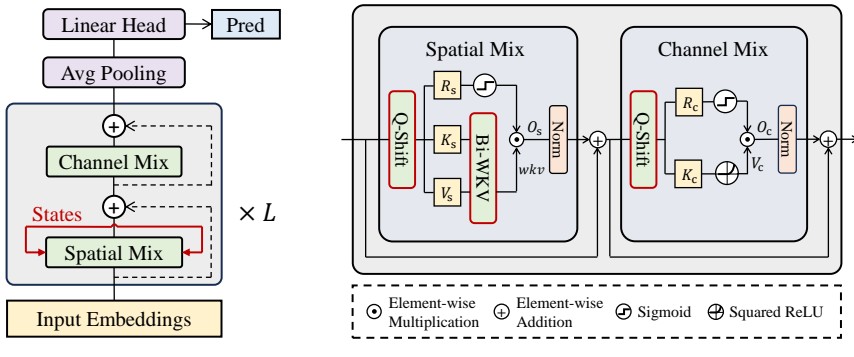

Figure 2: **Overall architecture of VRWKV.** (a) The VRWKV architecture includes $L$ identical VRWKV encoder layers, an average pooling layer, and a linear prediction head. (b) The details of the VRWKV encoder layer. Q-Shift denotes the quad-directional shift method tailed for vision tasks. The "Bi-WKV" module served as a bidirectional RNN cell or a global attention mechanism.

architecture (Vaswani et al., 2017) marked a turning point, with both NLP and computer vision fields shifting focus toward attention-based feature aggregation. The global attention mechanism overcomes the limitations of CNNs in modeling long-range dependencies and the shortcomings of RNNs in parallel computation while coming at a high computational cost.

To address these issues, researchers have introduced innovations such as window attention and spatial reduction attention. Window attention (Liu et al., 2021; Vaswani et al., 2021; Dai et al., 2022) restricts the self-attention computation within local windows, drastically reducing the computational complexity while preserving the receptive field through window-level interaction. Spatial reduction attention (Wang et al., 2021; 2022), on the other hand, reduces the dimensionality of the feature space before applying the attention mechanism, effectively decreasing the computational requirements without significantly degrading the model's performance.

In addition to the efforts to optimize the global attention mechanism, various operators with linear complexity have also been explored. For instance, RWKV (Peng et al., 2023) and RetNet (Sun et al., 2023) employed exponential decay to model global information efficiently while SSMs (Gu et al., 2021a;b; Smith et al., 2022a; Wang et al., 2023a) also exhibited linear complexity concerning sequence length and modification in Mamba (Gu & Dao, 2023) enable them to be input-dependent. Besides, XCA (Ali et al., 2021) achieved linear complexity by calculating the cross-variance between input tokens. However, the low efficiency of information interaction between tokens makes the need for the assistance of additional modules necessary to complete a comprehensive feature aggregation. Despite some concurrent efforts (Liu et al., 2024; Zhu et al., 2024; Fan et al., 2023), adapting these NLP-derived techniques to vision tasks remains a challenge in maintaining stable training across larger and more complex vision models.

## 3 VISION-RWKV

### 3.1 OVERALL ARCHITECTURE

In this section, we propose Vision-RWKV (VRWKV), an efficient vision encoder with a linear complexity attention mechanism. Our principle is to preserve the advantages of the original RWKV architecture (Peng et al., 2023), making necessary modifications to enable its flexible application in vision tasks, supporting sparse input, and ensuring the stability of the training process after scaling up. An overview of our VRWKV is depicted in Figure 2.

VRWKV adopts a block-stacked image encoder design like ViT, where each block consists of a spatial-mix module and a channel-mix module. The spatial-mix module functions as an attention mechanism, performing linear complexity global attention computation while the channel mix module serves as a feed-forward network (FFN), performing feature fusion in the channel dimension. The entire VRWKV includes a patch embedding layer and a stack of $L$ identical VRWKV encoder layers, where each layer maintains the input resolution.

**Data Flow.** First, we transform the $H \times W \times 3$ image into $HW/p^2$ patches, where $p$ denotes the patch size. The patches after a linear projection add the position embedding to obtain image tokens of shape $T \times C$, where $T = HW/p^2$ denotes the total number of tokens. These tokens are then input into the VRWKV encoder with $L$ layers.

In each layer, tokens are first fed into the spatial-mix module which plays the role of a global attention mechanism. Specifically, as shown in Figure 2(b), the input tokens are first shifted and fed into three parallel linear layers to obtain the matrices $R_s, K_s, V_s \in \mathbb{R}^{T \times C}$:

$$R_\text{s} = \text{Q-Shift}_R(X)W_R, \quad K_\text{s} = \text{Q-Shift}_K(X)W_K, \quad V_\text{s} = \text{Q-Shift}_V(X)W_V. \tag{1}$$

Here, the Q-Shift operator is a token shift function specially designed for the information exchange through nearby tokens according to the visual prior. $K_\text{s}$ and $V_\text{s}$ are then passed to calculate the global attention result, $wkv \in \mathbb{R}^{T \times C}$, by a linear complexity bidirectional attention mechanism, Bi-WKV, and multiplied with $\sigma(R)$ which controls the output $O_\text{s}$ probability:

$$O_\text{s} = (\sigma(R_\text{s}) \odot wkv)W_O,$$
$$\text{where } wkv = \text{Bi-WKV}(K_\text{s}, V_\text{s}). \tag{2}$$

Operator $\sigma$ denotes the sigmoid function, and $\odot$ represents element-wise multiplication. The output features are then stabilized using layer normalization (Ba et al., 2016) following the linear projection.

Subsequently, the tokens are passed into the channel-mix module for a channel-wise fusion. $R_\text{c}, K_\text{c}$ are obtained in a similar manner as spatial-mix:

$$R_\text{c} = \text{Q-Shift}_R(X)W_R, \quad K_\text{c} = \text{Q-Shift}_K(X)W_K. \tag{3}$$

In the channel-mix module, $V_\text{c}$ is a linear projection of $K_\text{c}$ after the activation function and controlled by a gate mechanism $\sigma(R_\text{c})$. The output $O_\text{c}$ is the linear projection of the aforementioned result:

$$O_\text{c} = (\sigma(R_\text{c}) \odot V_\text{c})W_O,$$
$$\text{where } V_\text{c} = \text{SquaredReLU}(K_\text{c})W_V. \tag{4}$$

Simultaneously, residual connections (He et al., 2016) are established from the tokens to each normalization layer to ensure that training gradients do not vanish in deep networks.

## 3.2 LINEAR COMPLEXITY BIDIRECTIONAL ATTENTION

Different from the vanilla RWKV (Peng et al., 2023), we make the following modifications to its original attention mechanism: (1) **Bidirectional attention**: We extend the upper limit of original RWKV attention from $t$ (the current token) to $T - 1$ (the last token) to ensure that all tokens are mutually visible in the calculation of each result. Thus, the original causal attention transforms into bidirectional global attention. (2) **Relative bias**: We compute the absolute value of the time difference $t - i$ and divide it by the total number of tokens (denoted as $T$) to represent the relative bias of tokens in images of different sizes. (3) **Flexible decay:** We no longer restrict the learnable decay parameter $w$ to be positive in the exponential term allowing the exponential decay attention to focus on tokens further away from the current token.

Under the collective influence of these ingenious modifications, we achieve global attention while maintaining linear complexity to the input token number $T$, thereby maximizing the preservation of RWKV's inherent low complexity and extending it to the visual domain.

Similar to RWKV, our bidirectional attention can also be equivalently expressed in a summation form (for the sake of clarity) and an RNN form (in practical implementation).

**Summation Form.** The attention calculation result for the $t$-th token is given by the formula:

$$wkv_t = \text{Bi-WKV}(K, V)_t = \frac{\sum_{i=0, i \neq t}^{T-1} e^{-(|t-i|-1)/T \cdot w + k_i} v_i + e^{u+k_t} v_t}{\sum_{i=0, i \neq t}^{T-1} e^{-(|t-i|-1)/T \cdot w + k_i} + e^{u+k_t}}. \tag{5}$$

Here, $T$ represents the total number of tokens, equal to $HW/p^2$, $w$ and $u$ are two $C$-dimensional learnable vectors that represent channel-wise spatial decay and the bonus indicating the current token, respectively. $k_t$ and $v_t$ denotes $t$-th feature of $K$ and $V$.

The summation formula indicates that the output $wkv_t$ is a weighted sum of $V$ along the token dimension from 0 to $T - 1$, resulting in a $C$-dimensional vector. It represents the result obtained

| Model | Emb Dim | Hidden Dim | Depth | Extra Norm | #Param |
|-------|---------|-----------|-------|-----------|--------|
| VRWKV-T | 192 | 768 | 12 | ✗ | 6.2M |
| VRWKV-S | 384 | 1536 | 12 | ✗ | 23.8M |
| VRWKV-B | 768 | 3072 | 12 | ✗ | 93.7M |
| VRWKV-L | 1024 | 4096 | 24 | ✓ | 334.9M |

Table 1: **Default settings for Vision-RWKV of different scales.** We report the embedding dimension, hidden dimension, and model depth. "Extra Norm" means additional layer normalization layers are used to stabilize the model's outputs. "#Param" denotes the number of parameters.

by applying the attention operation to the t-th token. The weight is determined by the spatial decay vector $w$, the relative bias between tokens $(|t - i| - 1)/T$, and $k_i$ collectively.

**RNN Form.** In the practical implementation, the above Eq 5 can be transformed into a recursive formula in the form of RNN that the result of each token can be obtained through a fixed number of FLOPs. By splitting the summation term of the denominator and numerator in Eq 5 with $t$ as the boundary, we can obtain 4 hidden states:

$$a_{t-1} = \sum_{i=0}^{t-1} e^{-(|t-i|-1)/T \cdot w + k_i} v_i, \quad b_{t-1} = \sum_{i=t+1}^{T-1} e^{-(|t-i|-1)/T \cdot w + k_i} v_i,$$

$$c_{t-1} = \sum_{i=0}^{t-1} e^{-(|t-i|-1)/T \cdot w + k_i}, \quad d_{t-1} = \sum_{i=t+1}^{T-1} e^{-(|t-i|-1)/T \cdot w + k_i}, \tag{6}$$

that can be recursively computed due to its mathematical formulation. The update of hidden states only requires adding or subtracting one summation term and multiplying or dividing $e^{-w/T}$. Then the $t$-th result can be given as:

$$wkv_t = \frac{a_{t-1} + b_{t-1} + e^{k_t + u} v_t}{c_{t-1} + d_{t-1} + e^{k_t + u}}. \tag{7}$$

Each update step yields an attention result (*i.e.*, $wkv_t$) for a token, so the entire $wkv$ matrix requires $T$ steps.

When the input $K$ and $V$ are matrices with the shape of $T \times C$, the computational cost of calculating the $wkv$ matrix is given by:

$$\text{FLOPs(Bi-WKV}(K, V)) = 13 \times T \times C. \tag{8}$$

Here, the number 13 is approximately from the updates of $(a, b, c, d)$, the computation of the exponential, and the calculation of $wkv_t$. The above approximation shows that the complexity of the forward process is $O(TC)$. The backward propagation of the operator can still be represented as a more complex RNN form, with a computational complexity of $O(TC)$. The specific formula for forward updating and backward propagation is provided in the Appendix A.1.

## 3.3 QUAD-DIRECTIONAL TOKEN SHIFT

We introduce a quad-directional token shift (Q-Shift) as a flexible extension of the original token shift operation in RWKV in the first step of each spatial-mix and channel-mix module. The Q-Shift operation allows all tokens shifted and linearly interpolated with their neighboring tokens as follows:

$$\text{Q-Shift}_{(*)}(X) = X + (1 - \mu_{(*)})X^\dagger,$$

$$\text{where } X^\dagger[h, w] = \text{Concat}(X[h-1, w, 0:C/4], X[h+1, w, C/4:C/2], \tag{9}$$

$$X[h, w-1, C/2:3C/4], X[h, w+1, 3C/4:C]).$$

Subscript $(*) \in \{R, K, V\}$ denotes 3 interpolation of $X$ and $X^\dagger$ controlled by the learnable vectors $\mu_{(*)}$ for the later calculation of $R, K, V$, respectively. $h$ and $w$ denote the row and column index of token $X$, ":" is a slicing operation excluded the end index. The Q-Shift makes the attention mechanism of different channels obtain the prior of focusing on neighboring tokens internally without introducing many additional FLOPs. It also increases the receptive field of each token which greatly enhances the coverage of the token in the posterior layers. $X^\dagger$ is obtained by slicing X without introducing new computations, allowing for flexible transformations during the training process for different tasks. When handling sparse inputs that do not contain original image space information, such as masked image modeling, shifting can be applied in a single dimension to maximize the preservation of image priors.

| | Method | Size | #Param | FLOPs | Top-1 Acc |
|---|---|---|---|---|---|
| hierarchical | ResNet-18 (He et al., 2016) | $224^2$ | 11.7M | 1.8G | 69.9 |
| | PVT-T (Wang et al., 2021) | $224^2$ | 13.2M | 1.9G | 75.1 |
| | ResNet-50 (He et al., 2016) | $224^2$ | 25.6M | 4.1G | 76.6 |
| | Swin-T (Liu et al., 2021) | $224^2$ | 28.3M | 4.4G | 81.2 |
| | PVT-M (Wang et al., 2021) | $224^2$ | 44.2M | 6.7G | 81.2 |
| | ResNet-101 (He et al., 2016) | $224^2$ | 44.6M | 7.9G | 78.0 |
| | Swin-S (Liu et al., 2021) | $224^2$ | 49.6M | 8.7G | 83.0 |
| | PVT-L (Wang et al., 2021) | $224^2$ | 61.4M | 9.8G | 81.7 |
| | Swin-B (Liu et al., 2021) | $224^2$ | 87.8M | 15.1G | 83.4 |
| non-hierarchical | DeiT-T (Touvron et al., 2021a) | $224^2$ | 5.7M | 1.3G | 72.2 |
| | DeiT-S (Touvron et al., 2021a) | $224^2$ | 22.1M | 4.6G | 79.9 |
| | XCiT-S12 (Ali et al., 2021) | $224^2$ | 26.0M | 4.8G | 82.0 |
| | DeiT-B (Touvron et al., 2021a) | $224^2$ | 86.6M | 17.6G | 81.8 |
| | XCiT-L24 (Ali et al., 2021) | $224^2$ | 189.0M | 36.1G | 82.9 |
| | ViT-L (Dosovitskiy et al., 2020) | $384^2$ | 309.5M | 191.1G | 85.2 |
| | VRWKV-T | $224^2$ | 6.2M | 1.2G | 75.1 |
| | VRWKV-S | $224^2$ | 23.8M | 4.6G | 80.1 |
| | VRWKV-B | $224^2$ | 93.7M | 18.2G | 82.0 |
| | VRWKV-L | $384^2$ | 334.9M | 189.5G | 86.0 |
| | VRWKV-L$^\dagger$ | $384^2$ | 334.9M | 189.5G | 86.2 |
| | VRWKV-L$^\star$ | $384^2$ | 334.9M | 189.5G | 86.5 |

Table 2: **Validation results on ImageNet-1K.** VRWKV-T/S/B are trained on ImageNet-1K, while VRWKV-L is pre-trained on Imagenet-22K and fine-tuned on Imagenet-1K. "#Param" denotes the number of parameters, and "FLOPs" represents the computational workload for the specified image resolution in the "Size" column. "$\dagger$" means additional MAE pre-training is applied in the pre-training process. "$\star$" indicates Bamboo-47K is used in the pre-training.

### 3.4 SCALE UP STABILITY

Increasing model depth and the accumulation of exponential terms during recursion can lead to instability in the training process. To mitigate this, we propose two simple yet effective adjustments: (1) **Bounded exponential**: As input resolution increases, both exponential decay and growth can quickly exceed the range of floating-point numbers. To address this, we divide the exponential term by the number of tokens (*e.g.*, $\exp(-(|t-i|-1)/T \cdot w)$), making the maximum decay and growth bounded. (2) **Extra layer normalization**: As models become deeper, we apply layer normalization (Ba et al., 2016) after the attention mechanism and the Squared ReLU operation, to prevent the model's output from overflowing. These two adjustments promote stable scaling of input resolution and model depth, facilitating the smooth convergence of large models. Additionally, we incorporate layer scale (Touvron et al., 2021b), which further enhances model stability during scaling.

### 3.5 MODEL DETAILS

Following ViT, the hyper-parameters for variants of VRWKV, including embedding dimension, hidden dimension in linear projection, and depth, are specified in Table 1. Due to the increased depth of the VRWKV-L model, additional layer normalizations as discussed in Section 3.4, are incorporated at appropriate positions to ensure output stability.

## 4 EXPERIMENTS

We comprehensively evaluate the substitutability of our VRWKV for ViT in performance, scalability, flexibility, and efficiency. The model's effectiveness is validated in image classification, object detection, and semantic segmentation tasks.

### 4.1 IMAGE CLASSIFICATION

**Settings.** For -Tiny/Small/Base models, we conduct supervised training from scratch on ImageNet-1K (Deng et al., 2009). Following the training strategy and data augmentation of DeiT (Touvron et al., 2021a), we use a batch size of 1024, AdamW (Loshchilov & Hutter, 2017) with a base learning rate of 5e-4, weight decay of 0.05, and cosine annealing schedule (Loshchilov & Hutter, 2016). Images are cropped to the resolution of $224 \times 224$ for training and validation. For the -Large models,

| Method | Window | #Param | FLOPs | AP$^b$ | AP$^m$ |
|--------|--------|--------|-------|--------|--------|
| ViT-T (Touvron et al., 2021a) | ✓ | 8.0M | 95.4G | 41.1 | 37.5 |
| ViT-T (Touvron et al., 2021a) | ✗ | 8.0M | 147.1G | 41.6 | 37.9 |
| VRWKV-T (ours) | ✗ | 8.4M | 67.9G | 41.7 | 38.0 |
| ViT-S (Touvron et al., 2021a) | ✓ | 27.5M | 241.2G | 44.6 | 39.7 |
| ViT-S (Touvron et al., 2021a) | ✗ | 27.5M | 344.5G | 44.9 | 40.1 |
| VRWKV-S (ours) | ✗ | 29.3M | 189.9G | 44.8 | 40.2 |
| ViT-B (Touvron et al., 2021a) | ✓ | 99.5M | 686.7G | 46.2 | 41.5 |
| ViT-B (Touvron et al., 2021a) | ✗ | 99.5M | 893.3G | 46.8 | 41.8 |
| VRWKV-B (ours) | ✗ | 106.6M | 599.0G | 46.8 | 41.7 |
| ViT-L (Steiner et al., 2021) | ✓ | 327.0M | 1799.3G | 48.7 | 43.3 |
| VRWKV-L (ours) | ✗ | 351.9M | 1730.6G | 50.6 | 44.9 |

Table 3: **Object detection and instance segmentation on COCO val2017.** All models adopt the ViT-Adapter (Chen et al., 2023) to generate multi-scale features for detection heads. -T/S/B models are initialized with ImageNet-1K weights, and all -L models use ImageNet-22K weights. "#Param" indicates the backbone parameter, and 'FLOPs' represent the backbone's computational workload for a $1333 \times 800$ input. "Window" denotes the use of window operation in ViT layers.

we first pre-train them for 90 epochs on ImageNet-22K with a batch size of 4096 and resolution of $192 \times 192$, and then fine-tune them for 20 epochs on ImageNet-1K to a higher resolution of $384 \times 384$.

**Results.** We compare the results of our VRWKV with other hierarchical and non-hierarchical backbones on the ImageNet-1K validation dataset. As shown in Table 2, with the same number of parameters, computational complexity, and training/testing resolutions, VRWKV achieves better results than ViT. For example, when VRWKV-T has slightly lower FLOPs than DeiT-T(1.2G *vs* 1.3G), our VRWKV-T achieves a 2.9 point higher than DeiT-T on top-1 accuracy. When the model size scales up, VRWKV still demonstrates higher baseline performance. VRWKV-L achieves a 0.8 point higher top-1 accuracy of 86.0% at the resolution of $384 \times 384$ than ViT-L, with a slightly reduced computational cost. The superior performance from tiny to large-size models demonstrates that the VRWKV model possesses the scalability as ViT. Additionally, after using a larger dataset Bamboo-47K (Zhang et al., 2022) in the pre-train process, the performance of VRWKV-L can be further boosted to 86.5%, indicating that our VRWKV also possesses the ability like ViT to benefit from pre-training on large-scale datasets. The exploration of VRWKV in classification tasks demonstrates its potential to be a viable alternative to traditional ViT models.

## 4.2 OBJECT DETECTION

**Settings.** In the detection tasks, we adopt Mask R-CNN (He et al., 2017) as the detection head. For the -Tiny/Small/Base models, the backbones use weights pre-trained on ImageNet-1K for 300 epochs. For the -Large model, weights pre-trained on ImageNet-22K are used. All models use a $1\times$ training schedule (*i.e.*, 12 epochs) with a batch size of 16, and AdamW (Loshchilov & Hutter, 2017) optimizer with an initial learning rate of 1e-4 and weight decay of 0.05.

**Results.** In Table 3, we report the detection results on the COCO val (Lin et al., 2014) dataset using VRWKV and ViT as backbones. As the results showed in Figure 1(a) and Table 3, due to the use of window attention in dense prediction tasks, VRWKV with global attention can achieve better performance than ViT with lower FLOPs. For example, VRWKV-T has approximately 30% lower backbone FLOPs compared to ViT-T using window attention, with an improvement of AP$^b$ by 0.6 points. Additionally, we compare the performance of VRWKV and ViT using global attention. For instance, VRWKV-S achieves similar performance to ViT-S with 45% lower FLOPs. This demonstrates the effectiveness of VRWKV's global attention mechanism in dense prediction tasks and the advantage of lower computational complexity compared to the original attention mechanism.

## 4.3 SEMANTIC SEGMENTATION

**Settings.** In the semantic segmentation task, we use UperNet (Xiao et al., 2018) as the segmentation head. Specifically, all ViT models use global attention in the segmentation task. For the -Tiny/Small/Base models, the backbones use weights pre-trained on ImageNet-1K. And for the -Large model, weights pre-trained on ImageNet-22K are used. We employ the AdamW optimizer

| Method | Window | #Param | FLOPs | mIoU |
|---|:---:|---:|---:|---:|
| ViT-T (Touvron et al., 2021a) | ✕ | 8.0M | 20.9G | 42.6 |
| VRWKV-T (ours) | ✕ | 8.4M | 16.6G | 43.3 |
| ViT-S (Touvron et al., 2021a) | ✕ | 27.5M | 54.0G | 46.2 |
| VRWKV-S (ours) | ✕ | 29.3M | 46.3G | 47.2 |
| ViT-B (Touvron et al., 2021a) | ✕ | 99.5M | 157.9G | 48.8 |
| VRWKV-B (ours) | ✕ | 106.6M | 146.0G | 49.2 |
| ViT-L (Steiner et al., 2021) | ✕ | 327.0M | 446.8G | 53.4 |
| VRWKV-L (ours) | ✕ | 351.9M | 421.9G | 53.5 |

Table 4: **Semantic segmentation on the ADE20K val set.** All models used ViT-Adapter (Chen et al., 2023) for multi-scale feature generation and are trained with UperNet as the segmentation heads. For consistency in comparison, all -T/S/B models are initialized using ImageNet-1K pre-training, whereas -L models utilize ImageNet-22K pre-training. "#Param" refers to the number of parameters of the backbone. We report the FLOPs of backbones with the input size of $512 \times 512$.

with an initial learning rate of 6e-5 for the -Small/Base/Large models and 12e-5 for the -Tiny model, a batch size of 16, and a weight decay of 0.01. All models are trained for 160k iterations on the training set of the ADE20K dataset (Zhou et al., 2017).

**Results.** As shown in Table 4, when using UperNet for semantic segmentation, models based on VRWKV consistently outperform those based on ViT with global attention, while also being more efficient. For example, VRWKV-S achieves 1 point higher than ViT-S with a 14% FLOPs decrease. VRWKV-L creates a result of 53.5 mIoU, similar to ViT-L, while the computation of the backbone has a 25G FLOPs lower. These results demonstrate that our VRWKV backbones can extract better features for semantic segmentation compared to ViT backbones while also being more efficient, benefiting from the linear complexity attention mechanism.

## 4.4 ABLATION STUDY

**Settings.** We conduct ablation studies of the tiny-size VRWKV on ImageNet-1K (Deng et al., 2009) to validate the effectiveness of various key components like Q-Shift and Bi-WKV. The experimental settings are consistent with Section 4.1.

**Token Shift.** We compare three approaches: no token shift, the original shift method used in RWKV, and our proposed Q-Shift. As shown in Table 5, the variation in the shift method shows performance differences. Variant 1 without token shift leads to a poor performance of 71.5, which is 3.6 points lower than our model. Even with the use of our global attention, the model using the original token shift still has a 0.7-point gap with our model.

| Method | Token Shift | Bidirectional Attention | Top-1 Acc |
|---|:---:|:---:|:---:|
| RWKV | original | ✕ | 71.1 (-4.0) |
| Variant 1 | none | ✓ | 71.5 (-3.6) |
| Variant 2 | original | ✓ | 74.4 (-0.7) |
| Variant 3 | Q-Shift | ✕ | 72.8 (-2.3) |
| VRWKV-T | Q-Shift | ✓ | 75.1 |

Table 5: **Ablation on key components of the proposed VRWKV.** All models are trained from scratch on ImageNet-1K. "Original" refers to the token shift in RWKV (Peng et al., 2023), which mixes tokens in a single direction.

**Bidirectional Attention.** The bidirectional attention mechanism enables the model to achieve global attention while the original RWKV attention has a causal mask internally. The result of Variant 3 shows the global attention mechanism brings a 2.3 points increase in the top-1 accuracy.

**Effective Receptive Field (ERF).** We analyze the impact of different designs on the ERF of models based on (Ding et al., 2022) and visualize it in Figure 3(a). We visualize the ERF of the central pixel with an input size of 1024 × 1024. In Figure 3(a), "No Shift" represents the absence of the token shift method (Q-Shift), and "RWKV Attn" indicates using the original RWKV attention mechanism without our modifications for vision tasks. From the comparison in the figure, all models except the "RWKV Attn" one achieved global attention while the global capacity of the VRWKV-T model is better than that of ViT-T. Despite the assistance of Q-Shift, the central pixel in "RWKV Attn" still cannot attend to the pixels on the bottom of the image due to the large input resolution. The results

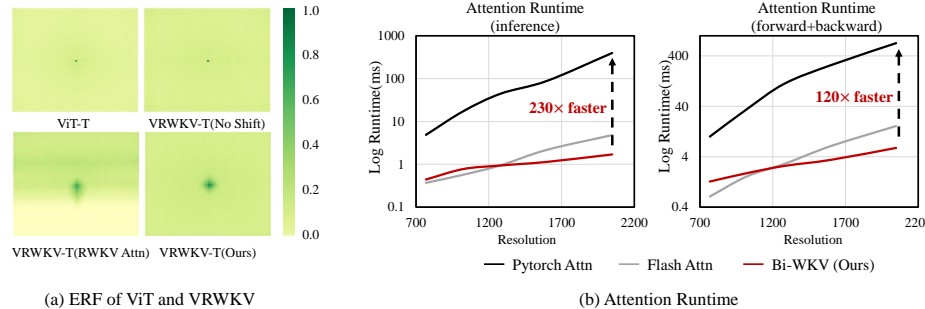

(a) ERF of ViT and VRWKV          (b) Attention Runtime

Figure 3: **Comparison of effective receptive field (ERF) and attention runtime.** (a) ERF for ViT and VRWKV in different settings. "No Shift" means no shift is used in spatial-mix and channel-mix modules. "RWKV Attn" means the original RWKV attention without our modifications. Our VRWKV with Q-Shift and Bi-WKV has a more comprehensive ERF than other counterparts. (b) Attention runtime of inference (left) and forward + backward (right) tested on an Nvidia A100 GPU.

of "No Shift" and Q-Shift show that the Q-Shift method expands the core range of the receptive field, enhancing the inductive bias of global attention.

**Efficiency Analysis.** To showcase the efficiency of our linear attention mechanism, we gradually increase the input resolution from $224 \times 224$ to $2048 \times 2048$ and compare the inference and memory efficiency of VRWKV-T and ViT-T. The results were tested on an Nvidia A100 GPU, as shown in Figure 1. From the curves presented in Figure 1(b), it is observed that at lower resolutions, VRWKV-T and ViT-T exhibit comparable memory usage and inference FPS. With the increase in input resolution, VRWKV-T shows a much higher speed than ViT-T. Additionally, VRWKV-T's RNN-like computational framework ensures a slow increase in GPU memory usage. By the time the resolution hits $2048 \times 2048$ (equivalent to 16384 tokens), VRWKV-T's inference speed is 10 times faster than ViT-T, and its memory consumption is reduced by 80% compared to ViT-T. It is worth mentioning that the implementation of the Q-Shift operation in PyTorch is highly inefficient, leading to an overall decrease in model speed. However, this operation can be optimized through other means (such as using CUDA extensions). Therefore, there is still potential for further improvement in the speed of VRWKV with better engineering optimization.

We also compare the speed of our attention mechanism kernel, Bi-WKV, with pytorch attention and flash attention (Dao et al., 2022), reported in Figure 3(b). Our Bi-WKV is significantly faster than the attention mechanism implemented using matrix multiplication (PyTorch attention), achieving a speedup of over a hundred times at a resolution of $2048 \times 2048$ (*i.e.*, 16384 tokens). Flash attention is highly optimized for memory I/O, and its matrix multiplication aligns well with the physical architecture of Nvidia GPUs, while our Bi-WKV lacks such hardware-level optimization. Nevertheless, in high-resolution scenarios, our Bi-WKV still demonstrates a significant speed advantage.

**MAE Pre-training.** ViTs can learn meaningful visual representations in masked image modeling (MIM). Yet, the rigor and effectiveness of linear attention vision models in this self-supervised pre-training paradigm have not been validated. Our VRWKV can handle sparse inputs and leverage MIM pre-training methods like MAE (He et al., 2017) by implementing a bidirectional shift operation that removes the vertical shift in Q-Shift. The pre-trained weights can be directly fine-tuned for other tasks using a Q-Shift manner. Following the same MAE pre-training setting as ViT, and subsequent classification training similar to Section 4.1, our VRWKV-L shows the ability to acquire visual prior from masked image modeling as shown in Table 2.

## 5 CONCLUSION

We propose Vision-RWKV (VRWKV), a vision encoder with a linear computational complexity attention mechanism. We demonstrate its capability to be an alternative backbone to ViT in comprehensive vision tasks including classification, dense predictions, and masked image modeling pre-training. With comparable performance and scalability, VRWKV exhibits lower computational complexity and memory consumption. Benefiting from its low complexity, VRWKV can achieve better performance in the tasks that ViT struggling to afford the high computational overhead of global attention. We hope VRWKV will be an efficient and low-cost alternative to ViT, showcasing the powerful potential of linear complexity transformers in vision fields.

## ACKNOWLEDGMENTS

This project was supported by the National Key R&D Program of China (No. 2022ZD0161300, 2022ZD0160101), the National Natural Science Foundation of China (No. 62376134). Zhe Chen is supported by the Youth PhD Student Research Project under the National Natural Science Foundation (No. 623B2050).

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

# A APPENDIX

## A.1 RNN FORM FORWARD AND BACKWARD

The attention mechanism in the spatial-mix module uses an RNN form forward and backward to achieve linear complexity of the input token number $T$. The following sections give more details of the operation.

| States | Recurrence Relation | Initial Value |
|---|---|---|
| $a$ | $a_t = \mathrm{w}^- \cdot a_{t-1} + e^{k_t} v_t$ | $a_{-1} = 0$ |
| $b$ | $b_t = \mathrm{w}^+ \cdot (b_{t-1} - e^{k_{t+1}} v_{t+1})$ | $b_{-1} = \sum_{i=1}^{T-1} e^{-(i-1)w+k_i} v_i$ |
| $c$ | $c_t = \mathrm{w}^- \cdot c_{t-1} + e^{k_t}$ | $c_{-1} = 0$ |
| $d$ | $d_t = \mathrm{w}^+ \cdot (d_{t-1} - e^{k_{t+1}})$ | $d_{-1} = \sum_{i=1}^{T-1} e^{-(i-1)w+k_i}$ |
| $\frac{da}{dw}$ | $\frac{da_t}{dw} = \mathrm{w}^- \cdot (\frac{da_{t-1}}{dw} - a_{t-1})$ | $\frac{da_{-1}}{dw} = 0$ |
| $\frac{db}{dw}$ | $\frac{db_t}{dw} = \mathrm{w}^+ \cdot (\frac{db_{t-1}}{dw} + b_{t-1} - e^{k_{t+1}} v_{t+1})$ | $\frac{db_{-1}}{dw} = \sum_{i=1}^{T-1} -(i-1)e^{-(i-1)w+k_i} v_i$ |
| $\frac{dc}{dw}$ | $\frac{dc_t}{dw} = \mathrm{w}^- \cdot (\frac{dc_{t-1}}{dw} - c_{t-1})$ | $\frac{dc_{-1}}{dw} = 0$ |
| $\frac{dd}{dw}$ | $\frac{dd_t}{dw} = \mathrm{w}^+ \cdot (\frac{dd_{t-1}}{dw} + d_{t-1} - e^{k_{t+1}})$ | $\frac{dd_{-1}}{dw} = \sum_{i=1}^{T-1} -(i-1)e^{-(i-1)w+k_i}$ |
| $ga$ | $ga_t = \mathrm{w}^+ \cdot (ga_{t-1} - gy_t \cdot y_t^{\text{iden}})$ | $ga_0 = \sum_{i=1}^{T-1} gy_i \cdot y_i^{\text{iden}} \cdot e^{-(i-1)w}$ |
| $gb$ | $gb_t = \mathrm{w}^- \cdot gb_{t-1} + gy_{t-1} \cdot y_{t-1}^{\text{iden}}$ | $gb_0 = 0$ |
| $gc$ | $gc_t = \mathrm{w}^+ \cdot (gc_{t-1} - gy_t \cdot y_t^{\text{iden}} \cdot y_t)$ | $gc_0 = \sum_{i=1}^{T-1} gy_i \cdot y_i^{\text{iden}} \cdot y_i \cdot e^{-(i-1)w}$ |
| $gd$ | $gd_t = \mathrm{w}^- \cdot gd_{t-1} + gy_{t-1} \cdot y_{t-1}^{\text{iden}} \cdot y_{t-1}$ | $gd_0 = 0$ |

Table 6: **RNN states of Bi-WKV in forward and backward process.** The update in recurrence relations has a fixed number of FLOPs. $\mathrm{w}^-$ and $\mathrm{w}^+$ denotes the grow or decay vector $e^{w/T}$ and $e^{-w/T}$. The calculation of initial values is $O(TC)$ which does not affect the final complexity.

### A.1.1 BACKWARD EQUATION

The backward process acquires the gradient of output matrix $wkv$ (denotes as $y$) passed from the previous layer (denotes as $gy$) to calculate the gradient of each input. The saved inputs are vectors $w, u \in \mathbb{R}^C$, key and value matrices $K, V \in \mathbb{R}^{T \times C}$ (the batch dimension is omitted). The new input is the gradient $gy \in \mathbb{R}^{T \times C}$ provided by the backpropagation. The outputs include the gradients $gw, gu \in \mathbb{R}^C$, matrices $gK, gV \in \mathbb{R}^{T \times C}$ corresponding to the inputs, respectively. From the RNN form of the forward process, the backward can be represented in an RNN form with a linear complexity related to the token number $T$. Some intermediate variables are listed as follows:

$$y_t^{\text{num}} = a_{t-1} + b_{t-1} + e^{u+k_t} v_t, \ y_t^{\text{iden}} = 1/(c_{t-1} + d_{t-1} + e^{u+k_t}), \ y_t = y_t^{\text{num}} \cdot y_t^{\text{iden}}. \tag{10}$$

The outputs of backward propagation are listed as follows:

$$gw = \sum_{t=0}^{T-1} gy_t \cdot y_t^{\text{iden}} \left( \frac{da_{t-1}}{dw} + \frac{db_{t-1}}{dw} - y_t \left( \frac{dc_{t-1}}{dw} + \frac{dd_{t-1}}{dw} \right) \right), \tag{11}$$

$$gu = \sum_{t=0}^{T-1} gy_t \cdot y_t^{\text{iden}} \cdot e^{u+k_t} (-y_t + v_t), \tag{12}$$

$$gk_t = gb_t \cdot e^{k_t} v_t - gd_t \cdot e^{k_t} + gy_t \cdot y_t^{\text{iden}} (e^{k_t+u} v_t - y_t \cdot e^{k_t+u})$$
$$+ ga_t \cdot e^{k_t} v_t - gc_t \cdot e^{k_t}, \tag{13}$$

$$gv_t = gb_t \cdot e^{k_t} + ga_t \cdot e^{k_t} + gy_t \cdot y_t^{\text{iden}} \cdot e^{k_t+u}. \tag{14}$$

The RNN states and their initial values and recurrence relations are provided in Table 6. From the recurrence relations, all updates have a complexity of $O(C)$, which means the number of FLOPs for each update is fixed. Therefore, the final backward complexity is $O(sTC)$ where s denotes the sum of the FLOPs for all equations.

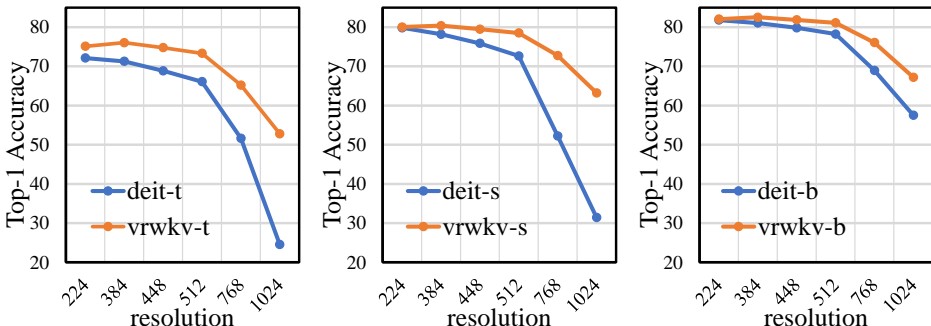

Figure 4: **Performance of VRWKV and DeiT (Touvron et al., 2021a) on ImageNet-1K (Deng et al., 2009).** All models are trained on a fixed resolution of $224 \times 224$ and evaluated on different resolutions. Our VRWKV shows an obvious robustness on different resolutions.

| Method | #Param | Top-1 Acc |
|---|---|---|
| Vim-T | 7M | 76.1 |
| VRWKV-T | 6M | 75.1 |
| Vim-S | 26M | 80.5 |
| VRWKV-S | 24M | 80.1 |
| Vim-B | 98M | 81.9 |
| VRWKV-B | 94M | 82.0 |
| Vim-L | NA | NA |
| VRWKV-L | 335M | 86.0 |

Table 7: **Comparison with Vision Mamba (Zhu et al., 2024)(Vim) on ImageNet-1K (Deng et al., 2009).** "NA" denotes not available.

### A.1.2 IMPLEMENTATION DETAILS

A numerical trick to compute safe exponential in (Peng et al., 2023; Milakov & Gimelshein, 2018) is used to avoid overflow in the exponential terms of the recurrence in the forward and backward process. An example of the update of state $a$ is shown as follows:

$$q := \max(p_{t-1} - w/T, k_t), \tag{15}$$
$$a' = \exp(-w/T + p_{t-1} - q) \cdot a' + \exp(k_t - q) \cdot v_t, \tag{16}$$
$$p_t = q. \tag{17}$$

The exponential terms in the new state $a'$ are forced to be smaller than 1 by subtracting the max value. The subtracted part stored in $p$ is divided automatically when calculating $wkv$.

### A.2 ROBUSTNESS ON IMAGE RESOLUTION

**Settings.** In this experiment, we aim to explore whether the proposed VRWKV exhibits distinct properties compared to ViT. To this end, we evaluated the performance of different variants of DeiT (Touvron et al., 2021a) and VRWKV at different resolutions on the ImageNet-1K (Deng et al., 2009) classification task. While the training was standardized at a resolution of $224 \times 224$, we evaluated the models across a range of resolutions, from $224 \times 224$ to $1024 \times 1024$.

**Results.** As shown in Figure 4, our VRWKV demonstrates stronger robustness when evaluated on a higher resolution. In contrast to DeiT (Touvron et al., 2021a), VRWKV performs better as the resolution slightly increases. For example, VRWKV-B achieved a top-1 accuracy of 82.5% at a $384 \times 384$ resolution, marking an improvement of 0.5 points over its accuracy at the training resolution. When the test resolution scales up to $1024 \times 1024$, VRWKV-B still maintains an accuracy of 67.2%, while DeiT-B only achieves an accuracy of 57.5%. This indicates that VRWKV has stronger potential and robustness in high-resolution scenarios and is a good alternative to ViT for high-resolution tasks.

### A.3 Comparison to Vision Mamba

In the field of visual linear attention mechanisms, Vision Mamba (Zhu et al., 2024)(Vim) stands out as a significant implementation that has garnered considerable interest. In this study, we compare the performance and efficiency of two models in visual tasks.

**Classification Performance.** We compare the classification accuracy on ImageNet-1K (Deng et al., 2009). As reported in Table 7, Vim has higher Top-1 Acc in tiny and small sizes, while the base size models achieve comparable performance. Benefitting from our careful stability design, VRWKV can scale up to larger models while Vim faces instability issues in the training process.

**Inference Efficiency.** We compare the inference speed of three attention mechanisms: Vanilla Attn (Vaswani et al., 2017), Bi-WKV, and Vision Mamba, shown in Figure 5. As the input resolution increases, the inference cost of vanilla attention quickly surpasses that of Bi-WKV and Vision Mamba. With our optimizations and design on CUDA, our Bi-WKV demonstrates faster speeds than Vision Mamba at the same input resolution.

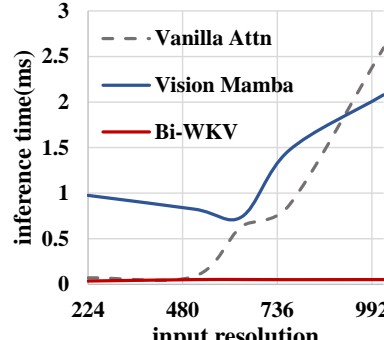

Figure 5: **Inference time of attention mechanisms.** Input resolutions are scanned from 224 to 1024. All experiments are run on Nvidia A100.

