# OpenReview forum: "Vision-RWKV: Efficient and Scalable Visual Perception with RWKV-Like Architectures"
_ICLR.cc/2025/Conference — ICLR 2025 Spotlight_

### Official Review · Reviewer_weiY · 2024-10-29

**Soundness:** 3
**Presentation:** 3
**Contribution:** 3
**Rating:** 8
**Confidence:** 3

**Summary:**

The authors describe a Vision-RWKV architecture that employs novel techniques like bi-directional RWKV, quad token shifting method Q-shift etc. These techniques helps Vision-RWKV  architecture surpass window-based ViTs and comparable to global attention ViTs along with lower FLOPs, lower GPU memory cost and faster processing as shown in Figure 1. Although the MAE finetuning is not as straightforward as typical finetuning on downstream task, overall this paper has a good contribution for the vision research community.

**Strengths:**

- Novel contribution for quad-directional token shifting called Q-shift. This essentially increases the models range of semantic understanding
- Authors expanded causal RWKV to a bidirectional global RWKV. They modified the exponent in the RWKV attention that leads to transforming the absolution positional bias to relative bias.
- Well written paper with through experimentation including MAE pretraining

**Weaknesses:**

- The authors implemented a bidirectional shift operation that removed the vertical shift in Q-shift, thereby enabling for MAE pretraining. IMO this is a source of complexity. As a result, MAE finetuning needs to be done in Q-shift manner

**Questions:**

- Can you please clarify what you meant in line 527-528 by task fine-tuning using Q-shift manner?

---

> ### Author Response · Authors · 2024-11-20
> **Response to Reviewer weiY**
>
> Thank you for your positive feedback on our work!
>
> **Q1: Why do we use Q-Shift in finetuning?**
>
> **A1:** Regarding the technical details, 75% of visual tokens are randomly masked (dropped) during the encoding process in MAE pre-training, resulting in the loss of 2D shape information for the image tokens. Therefore, methods like Q-Shift which require positional relationships from surrounding tokens are not available, so the shifting can only be performed in one direction. During the fine-tuning process, the substitution of the shift method with Q-Shift enables a more extensive exchange of local information. The shift operation involves only slice operations without learnable parameters, hence replacing the shift function is adequate and does not introduce additional complexity.

---

### Official Review · Reviewer_cUXV · 2024-11-02

**Soundness:** 3
**Presentation:** 4
**Contribution:** 2
**Rating:** 8
**Confidence:** 3

**Summary:**

Authors propose an approach that adapts NLP models for RWKV to vision tasks, termed as V-RWKV. Their proposed method could be looked upon as a cost effective solution than ViTs. They propose Quad-directional Shift and change the causal attention to bidirectional one towards learning locl concepts that are more relevant in vision rather than simple text. They evaluate their approach on the image classification, object detection and semantic segmentation tasks.

**Strengths:**

- Comprehensive results across three tasks.
- Interesting direction to replace ViTs with other efficient techniques that provide on-par or better performance.

**Weaknesses:**

- The gains in efficiency seem to be relatively minor, i.e., it is not an order of magnitude which still brings the question whether it is worth exploring these type of models to replace ViTs to begin with or not.

 For example, in Table 2 ViT-L vs. VRWKV-L 191.1 vs 189.5G FLOPS and parameters at 309.5M vs. 334.9M respectively. I think the reduction in FLOPS when scaling to Large variant seems to be around 3G. Similarly in Table 4, FLOPS 446.8 vs. 421.9G with the expense of an increase in the number of parameters. I am not expert in such type of methods focused on improving efficiency but I do not see the results are impressive enough to show the benefit from the V-RWKV design, especially that it is increasing the parameters.

**Questions:**

Clarifying the practical benefit from their proposed approach and why not simply use vanilla ViTs considering the current gains are not that considerable when looking at Tables 2 and 4. Hence, the reason I am leaning towards a marginal reject but since the method seems to provide interesting direction and their results for detection seems sufficiently good I am not going lower.

---

> ### Author Response · Authors · 2024-11-20
> **Response to Reviewer cUXV**
>
> Thanks for your constructive comments. We provide our feedback as follows.
>
> **Q1: Advantages in efficiency compared to ViT.**
>
> **A1:** At a low-resolution scenario, the primary factor of the total computational cost is from the linear projections in MLPs. Attention modules have low FLOPs, constituting a low proportion of the entire model. Therefore, the efficiency improvement of VRWKV compared to ViT is not significant in Table 2 (224\*224) and Table 4 (512\*512).
>
> At higher resolutions, VRWKV shows a noticeable reduction in FLOPs compared to ViT using full attention. **Such as in Table 3 where the detection resolution is 1333\*800, VRWKV-T has 50% lower FLOPs than ViT-T.**
>
> High resolution are common and important in many visual tasks, such as dense prediction tasks in Remote Sensing. Global attention mechanism is proven to play a crucial role in model performance\[1\]. In such scenarios, the application of VRWKV with linear complexity will offer greater efficiency advantages over ViT.
>
> [1] Zhao, S., Chen, H., Zhang, X., Xiao, P., Bai, L., & Ouyang, W. (2024). Rs-mamba for large remote sensing image dense prediction. arXiv preprint arXiv:2404.02668.

---

> ### Comment · Reviewer_cUXV · 2024-11-25
>
> Thanks for the authors for clarifying the relation to image resolution. Would it be possible to assess the performance of VRWKV improvement then on a semantic segmentation dataset with high resolution to validate this? I am just looking to ensure the generality of their method improving on multiple tasks to make it sufficient as an interesting work to replace vanilla ViTs. Since one task is not sufficient, so even one result evaluating their model on a semantic segmentation dataset with high resolution compared to ViTs would confirm this further and ensure its not just results tied to the detection task.
>
> Example:
> [1] COCO-Stuff: Thing and Stuff Classes in Context. H. Caesar, J. Uijlings, V. Ferrari, In Computer Vision and Pattern Recognition (CVPR), 2018.
>
> Or other dataset of the authors choice where SOA methods are already comparing on high resolution. Or even simpler for time sake what would be the computational efficiency gain when just simply performing inference with higher resolution in the semantic segmentation task for both their model vs. ViTs. Basically an initial indication that this is confirmed on another task should be sufficient even if time doesn't permit to formally compare with SOA methods on a corresponding high resolution image dataset.
>
> Since the authors have not shown any response to this inquiry I stay at borderline reject since to the best of my knowledge there is no explanation how is it scaling w.r.t resolution in segmentation beyond detection to confirm that their method is efficient and general across tasks.  There might be some other factors beyond their contribution that resulted in this scaling in detection and authors did not bother to show otherwise.

---

> > ### Author Response · Authors · 2024-12-03
> > **Response to Reviewer cUXV**
> >
> > Thanks for your insights on the analysis of high-resolution segmentation tasks. We hope the following clarification can address your concerns.
> >
> > **Q1: Performance of high resolution on segmentation tasks.**
> >
> > **A1:** We conducted a performance comparison between our Vision-RWKV model and ViT (full attention) and ViT-win (window attention) on the high-resolution remote sensing segmentation dataset ISPRS Potsdam[1]. We employed a shorter training setting to obtain effective comparative conclusions on base-sized models quickly.
> >
> > **Settings.**
> > We utilized UperNet as the segmentation head. All model backbones were pretrained on ImageNet-1K. The AdamW optimizer was employed with an initial learning rate of 1e-4. We trained models at two resolutions, 1024 and 2048. For 1024 resolution, we trained for 20k iterations, and for 2048 resolution, we trained for 40k iterations.
> >
> > **Results.**
> > | Model     | Resolution | #P     | FLOPs   | mIoU |
> > |:----------|:----------:|-------:|--------:|:----:|
> > | ViT-win-B | 1024       | 99.5M  | 667.1G  | 78.4 |
> > | ViT-B     | 1024       | 99.5M  | 863.4G  | 78.8 |
> > | VRWKV-B   | 1024       | 106.6M | 584.1G  | 78.9 |
> > | ViT-win-B | 2048       | 99.5M  | 2275.6G | 77.9 |
> > | ViT-B     | 2048       | 99.5M  | OOM   | -  |
> > | VRWKV-B   | 2048       | 106.6M | 2336.5G | 78.6 |
> >
> > As shown in the table above, at 1024 resolution our VRWKV-B has a 78.9 mIoU which **outperformed ViT-win-B and performed comparably to ViT-B with full attention**. Additionally, our VRWKV-B exhibited **32% lower computational complexity than ViT-B**. At the 2048 resolution, ViT-win-B's use of windowed attention led to performance degradation due to loss of global information, while training ViT-B with full attention resulted in out-of-memory(OOM) errors. Our VRWKV-B maintained performance consistent with that at 1024 resolution.
> >
> > These results underscore the superiority of VRWKV over ViT in terms of computational efficiency and global receptive field in high-resolution segmentation tasks.
> >
> > [1] 2D Semantic Labeling Contest - Potsdam. (n.d.). ISPRS. https://www.isprs.org/education/benchmarks/UrbanSemLab/2d-sem-label-potsdam.aspx

---

> > > ### Comment · Reviewer_cUXV · 2024-12-03
> > >
> > > Thanks for the authors on the results posted this has addressed indeed my main concern and it can be added in your supplementary material then.

---

### Official Review · Reviewer_QwNu · 2024-11-06

**Soundness:** 4
**Presentation:** 4
**Contribution:** 2
**Rating:** 8
**Confidence:** 2

**Summary:**

The paper introduces a new network architecture, VISION-RWKV, a vision-adapted version of the RWKV network from the NLP community, which employs an RNN-based linear attention mechanism. The authors have made necessary adaptations to the RWKV to better suit visual tasks, recognizing the differences between vision and language processing tasks.

**Strengths:**

1.The paper is well-written and easy to follow;

2.Extensive empirical evidence supports the effectiveness of the model, indicating its practical application potential;

3.The adaptation of RWKV for visual tasks goes beyond mere transfer, incorporating modifications that enhance its suitability for image processing.

**Weaknesses:**

Major：Although the paper discusses RWKV and mamba as prominent RNN-based linear attention models transitioning from NLP to computer vision, it lacks a direct comparison with the mamba (vision) model. Such a comparison would be valuable for assessing the respective strengths and weaknesses of each model in the field of computer vision.

Minor：From an innovation perspective, the approach of adapting other domains' mature architectures to computer vision, similar to what was done post-transformer, appears somewhat incremental.

**Questions:**

see weaknesses

---

> ### Author Response · Authors · 2024-11-20
> **Response to Reviewer QwNu**
>
> Thanks for your constructive comments. We provide our feedback as follows.
>
> **Q1: Comparison between Vision Mamba and Vision RWKV.**
>
> **A1:** Vision Mamba\[1\](Vim) and Vision RWKV (VRWKV) are both linear complexity models, but they differ in terms of model structure and scalability. We have added a comparison with Vim in Section A.3 in the Appendix.
>
> As demonstrated in Table 7, Vim exhibits superior Top-1 Accuracy in classification tasks for tiny and small sizes, whereas the base size models deliver similar performance. Benefitting from our careful stability design, VRWKV can scale up to larger models, while Vim encounters instability during training such as #issue30 in their official repository.
>
> When reproducing Vim-T for detection tasks, we encountered the same issue as described in #issue30, where the loss became NaN, and the final $AP_{box}$ was 25.5 before the training diverged. In contrast, **VRWKV did not exhibit any instability issues across models ranging from tiny to large** and achieved higher detection scores compared to ViT using window attention.
>
> Furthermore, we compare the inference speed of three attention mechanisms: Vanilla Attn, Bi-WKV, and Vision Mamba, shown in Figure 5 in the Appendix. As the input resolution increases, the inference cost of vanilla attention quickly surpasses that of Bi-WKV and Vision Mamba. With our optimizations and design on CUDA, our Bi-WKV demonstrates faster speeds than Vision Mamba at the same input resolution.
>
> **Q2: Innovation of Vision RWKV.**
>
> **A2:** As described in lines 46-53 of our paper, the direct application of the language model RWKV to the visual domain presents two significant challenges: (1) RWKV lacks a global receptive field in its attention mechanism, which is essential in the vision domain. (2) Ensuring the training stability of RWKV when the model scales up requires a more carefully designed approach.
>
> To address the first issue, we designed a bidirectional attention mechanism, Bi-WKV, which maintains linear complexity while achieving a global reception field. We also rewrote the operation at the CUDA level for better optimization. We also modified RWKV's shift method to better suit visual tasks.
> **In Table 5 presented in the ablation study, our implementation yielded a 4-point improvement in Top-1 Accuracy compared to directly employing the original RWKV model.**
>
> For the second issue, we have identified the appropriate key position to add normalization layers, making our model scale up stably while showing consistent improvement in model capacity.
>
> [1] Zhu, L., Liao, B., Zhang, Q., Wang, X., Liu, W., & Wang, X. (2024). Vision mamba: Efficient visual representation learning with bidirectional state space model. arXiv preprint arXiv:2401.09417.

---

> > ### Comment · Reviewer_QwNu · 2024-11-26
> >
> > Thank you to the authors for their response. Although the paper is not particularly outstanding in terms of innovation, I appreciate its engineering and practical applications, which are highly meaningful for the broader vision community. Therefore, I am inclined to accept this paper.

---

### Meta-Review · Area_Chair_VJcP · 2024-12-18

**Metareview:**

(a) Scientific Claims and Findings

The paper presents VISION-RWKV, an adaptation of the RWKV network from NLP to vision tasks, incorporating an RNN-based linear attention mechanism. Reviewer QwNu notes that the authors have made necessary modifications to suit visual tasks, while cUXV highlights the introduction of Quad-directional Shift and bidirectional attention to better capture local concepts in vision. weiY emphasizes the novel techniques like bi-directional RWKV and quad token shifting, which help the architecture surpass window-based ViTs in efficiency and performance.

(b) Strengths

The paper is well-written and provides extensive empirical evidence supporting the model's effectiveness, as noted by QwNu. cUXV appreciates the comprehensive results across multiple tasks and the exploration of efficient alternatives to ViTs. weiY highlights the novel contributions, such as the quad-directional token shifting and the expansion of causal RWKV to a bidirectional global RWKV, which enhance the model's semantic understanding and efficiency.

(c) Weaknesses

A major weakness identified by QwNu is the lack of direct comparison with the mamba (vision) model, which would help assess the model's strengths and weaknesses in computer vision. cUXV points out that the efficiency gains are relatively minor, questioning whether the model is a worthwhile replacement for ViTs given the modest reduction in FLOPs and increase in parameters. weiY mentions the complexity introduced by the bidirectional shift operation, which complicates MAE finetuning.

(d) Decision Reasons

The decision to accept the paper is based on its novel contributions to adapting NLP models for vision tasks and the comprehensive empirical validation provided. Despite some concerns about the incremental nature of the approach and the modest efficiency gains, the paper offers a promising direction for efficient vision models, as highlighted by all reviewers. The strengths in presentation, empirical results, and novel techniques outweigh the weaknesses, leading to an overall positive assessment.

**Additional Comments On Reviewer Discussion:**

During the rebuttal period, the authors addressed several concerns raised by the reviewers, leading to some adjustments in their evaluations.

Reviewer QwNu expressed appreciation for the authors' response, acknowledging that while the paper may not be groundbreaking in terms of innovation, its engineering and practical applications are valuable for the vision community. This led the reviewer to lean towards accepting the paper, emphasizing its practical significance.

Reviewer cUXV initially raised concerns about the generality of the method across different tasks, particularly questioning its performance on high-resolution semantic segmentation datasets. They suggested evaluating the model on such datasets to confirm its efficiency and general applicability beyond the detection task. The authors responded by providing results that addressed this concern, demonstrating the model's performance on a high-resolution semantic segmentation task. This additional evidence convinced cUXV of the method's broader applicability, leading them to increase their rating from a borderline reject to an accept, suggesting that these results could be included in the supplementary material.

In weighing these points for the final decision, the primary considerations were the practical applications and engineering contributions highlighted by QwNu, along with the additional validation provided by the authors in response to cUXV's concerns. The authors' ability to demonstrate the model's generality and efficiency across multiple tasks strengthened the case for acceptance, ultimately leading to a positive decision.

---

### Decision · Program_Chairs · 2025-01-22

Accept (Spotlight)